# A Four-Week Online Compassion and Gratitude Training Programme to Enhance Emotion Regulation: Implications for Stress Management and Healthcare Leadership

**DOI:** 10.3390/healthcare14010012

**Published:** 2025-12-20

**Authors:** Lotte Bock, Erik Riedel, Madiha Rana

**Affiliations:** 1Institute of Psychology in Education, Leuphana Universität Lüneburg, Universitätsallee 1, DE-21335 Lüneburg, Germany; 2Europäische Fernhochschule Hamburg, University of Applied Sciences, Doberaner Weg 20, DE-22143 Hamburg, Germany

**Keywords:** emotional intelligence, emotion regulation, mindfulness, compassion training, gratitude, psychological health, leadership, burnout prevention

## Abstract

**Background**: Emotional intelligence (EI), particularly the ability to regulate one’s emotions, is a key protective factor against stress and burnout in high-demand occupations, including leadership and healthcare. Compassion and gratitude practices have been proposed as brief, scalable methods to strengthen emotion regulation, yet empirical evidence from randomised controlled trials remains limited. **Objective**: This study evaluated whether a four-week, self-directed online programme combining daily loving-kindness meditation and gratitude journaling improves EI among leaders. **Methods**: Forty-five leaders in Germany from diverse occupational sectors were recruited via LinkedIn and Xing and were randomised using a computer-generated random sequence to an intervention or wait-list control group. EI was measured pre- and post-intervention with the Emotional Competence Questionnaire (EKF), comprising recognising one’s own feelings (RU), recognising others’ feelings (RO), regulating one’s own feelings (RC; primary outcome), and expressing feelings (RE). Adherence was reported in categorical form (e.g., daily, 3–5×/week, 1–2×/week). Treatment effects were tested using mixed-design ANOVAs. **Results**: A significant Group × Time interaction emerged for emotion regulation (RC), indicating greater improvement in the intervention group compared with the control group. No significant interaction effects were found for RU, RO, or RE. Adherence data did not permit dose–response analysis. **Conclusions**: A brief, self-directed online compassion and gratitude programme selectively improved emotion regulation—the EI facet most strongly linked to stress buffering and resilience. Although effects did not extend to other EI dimensions, findings suggest that low-threshold digital practices may strengthen a core emotional skill relevant to psychological well-being in leadership roles. Because the sample did not primarily comprise healthcare professionals, implications for healthcare settings re-main conceptual; targeted trials in clinical populations are warranted.

## 1. Introduction

The increasing prevalence of stress, burnout, and psychological strain in workplace and healthcare environments underscores the urgent need for accessible interventions to support mental health and well-being. Emotional intelligence (EI) has been identified as a protective factor against stress reactivity and psychological strain, particularly among leaders and healthcare professionals who work in high-pressure contexts [1,2,3].

Goleman’s influential book Emotional Intelligence: Why It Can Matter More Than IQ [4] popularised EI as a key determinant of success, highlighting its role in managing stress, improving communication, and fostering healthy interpersonal dynamics. EI enables individuals to comprehend and regulate their emotions and to respond effectively to the emotions of others, thereby supporting both psychological health and social functioning. Among the facets of EI, emotion regulation is most consistently associated with reduced stress reactivity, healthier coping strategies, and a lower risk of burnout [5,6,7].

Since Goleman’s work, evidence has accumulated showing that EI predicts leadership and occupational effectiveness [8,9], team performance [10], job satisfaction [11], and psychological well-being at work [12,13,14,15]. Improvements in emotion regulation are consistently found to be most proximal to stress buffering and mental health protection.

Mindful leadership has emerged as an approach integrating EI and mindfulness to enhance self-awareness, empathy, and emotional balance. By fostering presence and deliberate regulation, mindful leadership can reduce stress amplification within teams and improve resilience. These qualities are relevant for corporate leaders and are equally applicable to healthcare professionals, who routinely encounter high emotional demands, cognitive load, and risks of emotional exhaustion.

The present study evaluates whether a brief, self-directed online programme combining compassion and gratitude practices can strengthen EI—particularly emotion regulation—as a mechanism of stress management and psychological well-being. Our specific aim was to test whether a four-week compassion and gratitude programme produces greater improvements in EI among participants in the intervention group compared with a wait-list control group. We hypothesised that the intervention would improve all four EI facets, with emotion regulation specified a priori as the primary expected mechanism of change based on its established role in stress buffering and resilience.

## 2. Theoretical Framework and Hypotheses

The increasing prevalence of stress, burnout, and psychological strain in workplace and healthcare environments underscores the need for accessible interventions that can strengthen emotional resources. Emotional intelligence (EI) is recognised as a protective factor that supports adaptive coping, reduces stress reactivity, and promotes well-being in high-pressure roles. Among the facets of EI, emotion regulation is most directly tied to stress buffering and burnout prevention.

Compassion and gratitude practices represent two mindfulness-derived strategies that may enhance EI, particularly emotion regulation. Loving-kindness meditation fosters self-compassion, emotional balance, and resilience, whereas gratitude journaling has been linked to improved well-being and adaptive coping. These practices may therefore be especially relevant for leaders and healthcare professionals, who must navigate high emotional demands and regulate their affect effectively in daily work.

Mindful leadership integrates these capacities by promoting present-moment awareness, empathy, and deliberate emotional regulation. Strengthening emotion regulation through brief contemplative practices may therefore provide a mechanism for improving leaders’ psychological health and, conceptually, offer relevance for healthcare contexts where emotional resilience is critical.

The present study examines whether a four-week, self-directed online compassion and gratitude programme can improve EI—particularly emotion regulation—in leaders recruited from diverse occupational sectors in Germany. Participants engaged in daily loving-kindness meditation and gratitude journaling for approximately 20 min per day, with EI assessed at baseline and four weeks later.

Accordingly, our aims were to: (1) determine whether brief compassion and gratitude practices can enhance EI in a real-world leadership sample; (2) evaluate whether improvements differ between an intervention and wait-list control group; and (3) test emotion regulation (RC) as the primary mechanism of interest, given its established relevance for stress and well-being.

Reflecting Mayer and Salovey’s [16] model of EI and the theoretical role of emotion regulation in stress coping, we hypothesised that the experimental group would show greater pre–post improvements than the control group in:

**H****1.** 
*recognising and understanding their own emotions (RU);*


**H****2.** 
*recognising others’ emotions (RO);*


**H****3.** 
*regulating and controlling their own emotions (RC—primary outcome);*


**H****4.** 
*expressing emotions (RE).*


### 2.1. Mindful Leadership and Emotional Intelligence

Mindful leadership is acknowledged as an approach that promotes EI, empathy [17,18] self-compassion, resilience, and metacognition [7,19]. These qualities are frequently cited as protective factors for leaders’ psychological health and stress management. Recent work has linked mindful leadership to ethical decision-making, employee performance and well-being, and relationship quality [20,21,22,23,24], as well as to improvements in job performance, job satisfaction, need satisfaction, and reduced emotional exhaustion [7]. Mechanistically, these outcomes are often attributed to improved emotion regulation under pressure. Importantly, the same mechanisms are highly relevant for healthcare professionals, who face intense emotional demands, high cognitive load, and risks of burnout.

Mindful leadership and EI are interrelated, particularly around self-awareness, self-regulation, and empathy [25,26]. Mindful leadership fosters self-awareness by sustaining present-moment attention to thoughts, emotions, and bodily sensations. EI likewise entails accurate awareness of one’s feelings. Mindful leadership cultivates self-regulation through deliberate management of emotions—choosing considered responses rather than reacting impulsively—while EI similarly requires regulating impulses and affect. Mindful leadership also involves empathy through active listening and perspective-taking; EI emphasises recognising others’ emotions. Evidence indicates that emotion regulation is central to the positive association between mindfulness and EI [27]. Because emotion regulation downshifts stress reactivity and supports adaptive coping, it provides a plausible pathway from mindful leadership to leaders’ and healthcare workers’ psychological well-being.

Mindful leadership is increasingly recognised in organisations, yet relatively little research addresses how to develop these skills pragmatically. Compassion and gratitude training offer promising, low-burden methods to strengthen emotion regulation—the EI facet most directly tied to stress management—in time-constrained leadership and healthcare contexts.

### 2.2. Compassion and Gratitude Training

Compassion training draws on principles from various contemplative traditions, including mindfulness. Its primary aims are to cultivate self-compassion and compassion towards others, strengthen emotion regulation, and promote kind responses to challenging situations. Self-compassion, defined as extending kindness and understanding to oneself in moments of difficult [28,29], has been identified as a core construct underpinning such interventions. Deliberate practice of these skills has been shown to improve compassion towards others, receptivity to compassion from others, and self-compassion [30,31]. Additional benefits include enhanced emotional well-being, better interpersonal relationships, reduced stress reactivity, and greater prosocial behaviour [32]. Collectively, these outcomes position compassion training as a potential intervention to support leaders’ psychological health and stress management in demanding organisational contexts. Importantly, compassion training has also been identified as a protective factor for healthcare professionals, where it may buffer against emotional exhaustion, enhance patient care, and foster resilience in high-stress clinical environments.

Compassion training typically combines mindfulness practices that target emotion regulation with perspective shifting, cognitive restructuring, and prosocial behaviour exercises. A central element is loving-kindness meditation [33,34], which has demonstrated effects on positive affect and stress reduction. In both organisational and healthcare contexts, such practices may therefore serve as low-threshold, accessible approaches to strengthen emotional resources and promote well-being.

### 2.3. Loving-Kindness Meditation

Loving-kindness meditation, also known as metta meditation [35], is a contemplative practice in which practitioners systematically generate feelings of compassion and benevolence towards themselves, loved ones, strangers, and even difficult people. Although it originated in the Buddhist tradition, it has been adapted to modern contexts as a practical method for strengthening compassion. In organisational and leadership settings, it is often framed as a tool for cultivating resilience and stress-buffering emotional resources. In healthcare contexts, loving-kindness meditation has been highlighted as a promising approach to reducing burnout, enhancing empathy, and improving patient–provider relationships, given its ability to foster sustained compassion under stressful conditions.

Research shows that loving-kindness meditation has a particularly advantageous impact on compassion training [36]. A meta-analysis concluded that while the effectiveness of compassion training overall remains mixed, incorporating loving-kindness meditation is associated with improvements in life satisfaction [37]. Experimental studies similarly demonstrate that loving-kindness practice promotes resilience, enhances satisfaction and personal competence, and decreases depression, anxiety, and stress [38,39,40]. These findings align with stress-management outcomes highly relevant for leaders and healthcare professionals, including reduced negative affect and enhanced coping. Overall, loving-kindness meditation reflects key aspects of EI and mindful leadership, particularly emotion regulation and empathy, which support psychological well-being in both workplace and healthcare environments.

The meta-analysis by Gu et al. [37] noted that results remain promising but inconsistent, with further research needed. While nine moderators were considered, neither training format nor gratitude was included, despite evidence for gratitude’s positive workplace effects [41]. This gap provides a rationale for combining loving-kindness meditation with gratitude exercises in brief online interventions targeting stress management in leadership and healthcare contexts.

### 2.4. The Effect of Gratitude

Like compassion training, the impact of gratitude has attracted increasing interest over the past decade [42]. According to Bono & Sender [43], who reviewed a wide range of studies, gratitude can reduce antisocial behaviour, buffer stress, improve mental and physical health, strengthen interpersonal relationships, and reinforce resilience. Bono and Sender argue that gratitude fosters personal growth and enables individuals to inspire others, which aligns with the core tenets of mindful leadership as well as overlapping components of EI [44,45]. In leadership contexts, brief gratitude reflections—such as daily journaling—may therefore provide a low-burden strategy to support psychological well-being and stress regulation.

In healthcare contexts, gratitude has also been linked to reduced burnout, improved empathy, and higher job satisfaction among healthcare professionals, with evidence suggesting that cultivating gratitude can strengthen providers’ emotional resources and, indirectly, improve patient care outcomes. The practice of recording daily gratitude, for instance, has been shown to foster resilience and emotional balance in individuals exposed to chronic stressors, such as healthcare workers.

Nevertheless, Dickens’ [46] meta-analysis cautions that existing literature may overstate the benefits of gratitude. Further investigation is needed, particularly into whether combining gratitude with other contemplative practices, such as loving-kindness meditation, produces synergistic effects on emotion regulation and stress management.

### 2.5. The Format of Compassionate Training

Growing interest in the benefits of compassion training has led to the development of structured programmes such as Mindfulness-Based Compassion Training (MBCT), a counterpart to the well-established Mindfulness-Based Stress Reduction (MBSR) programme [47]. Both consist of eight weekly sessions, lasting between 2.5–3.5 h each, and can be delivered either in-person or via Virtual Instructor Led Training (VILT). Although MBCT and MBSR are distinct in aims and objectives, there remains a strong need to develop cost-effective, low-threshold training programmes, since conventional mindfulness programmes are often time-consuming and expensive.

Meta-analytic evidence suggests that online mindfulness training significantly improves mindfulness skills and reduces stress [48]. Such programmes offer flexibility and feasibility for both leaders and healthcare professionals, making them a low-threshold option to reach a wide audience. More recent findings demonstrate that mindfulness abilities can be enhanced in as little as four weeks via online training, with observable improvements in subjective stress perceptions [49]. For individuals managing heavy workloads and chronic stressors—including healthcare staff exposed to emotionally demanding patient interactions—such low-intensity yet effective interventions are especially relevant.

In summary, compassion training augments EI and mindful leadership by enhancing empathy, self-awareness, and emotion regulation. Its benefits can potentially be amplified when paired with gratitude practices. Beyond corporate contexts, these combined approaches may also reduce stress and foster resilience among healthcare professionals, where improved emotional regulation directly supports caregiver well-being and indirectly contributes to patient care quality. Nonetheless, it remains uncertain whether self-rated EI can be significantly improved through a brief, fully online four-week programme incorporating daily loving-kindness meditation and gratitude exercises. Testing this question is critical for determining whether such interventions can both strengthen EI and provide meaningful stress-management and health promotion benefits across occupational and healthcare populations.

## 3. Methodology

This study was prospectively registered with the German Clinical Trials Register (DRKS00030973) prior to participant enrolment, received approval from the Euro-FH Ethics Committee (EKEFH03/22), and obtained written informed consent from all participants. It was conducted as a randomised controlled trial with two assessment periods—baseline and post-intervention (February–March 2023)—to examine short-term changes in emotional intelligence (EI) facets relevant to leaders’ stress management and psychological health, in accordance with CONSORT recommendations (Figure 1).

Leaders were recruited in Germany through two professional social media platforms, LinkedIn and Xing, using two strategies: (a) posts in leadership-focused groups and (b) paid LinkedIn advertisements targeted at individuals in leadership roles. The paid advertising approach generated the majority of enrolments (approximately 70%), whereas group postings yielded fewer responses. Across all channels, an estimated 4000–5000 individuals were reached, resulting in 52 registrations.

Recruitment materials included a brief informational video (approximately 45 s), prepared by the first author, which outlined the study purpose, intervention components, expected daily time commitment, and participation requirements. Participants were employed across diverse occupational sectors (e.g., manufacturing, finance, technology, services, public sector). Recruitment was not restricted to healthcare settings, and most participants did not work in healthcare. The study therefore examines cross-occupational effects with conceptual relevance for healthcare, given the importance of emotion regulation for stress buffering and burnout prevention in clinical work. Sector was not used as a stratification variable due to the sample size.

Eligibility criteria required that participants (1) had held a leadership position in Germany for at least two years; (2) committed to approximately 20 min of daily practice during the four-week programme; and (3) agreed to complete both the baseline and post-intervention assessments. No additional exclusion criteria were applied.

Fifty-two leaders initially registered. To qualify, participants had to (1) have held a leadership position in Germany for at least two years; (2) commit to approximately 20 min per day of compassion and gratitude practice during the four-week programme; and (3) agree to complete two questionnaires, one at baseline and one post-intervention. Fifty leaders met these criteria, and 45 completed both surveys and were included in the analysis. After the baseline survey, participants were informed of their group allocation. Of the 50 randomised participants, five did not complete the post-intervention questionnaire and therefore had no outcome data available. As missingness occurred only at the post-assessment, an intention-to-treat analysis was not feasible. Accordingly, the primary analyses were conducted per-protocol, including the 45 participants with complete pre- and post-data. Baseline comparisons indicated no significant differences between completers and non-completers on demographic characteristics or baseline EI scores.

A priori power analysis (G*Power 3.1) indicated that a total sample size of 46 would be sufficient to detect a medium within-subject effect (d = 0.44) with α = 0.05 and β = 0.90. This calculation was based on paired pre–post comparisons and not on the between-group interaction. Because the achieved group sizes (*n* = 24 vs. *n* = 21) provide lower statistical power for detecting Group × Time interactions—particularly for small effects—the between-group analyses should be interpreted with this limitation in mind. Importantly, validated tools for precise a priori estimation of mixed-design ANOVA interaction power are limited; therefore, we report this constraint transparently as part of the study’s methodological considerations. The significant interaction for emotion regulation (RC) should thus be viewed as a robust effect despite conservative power conditions, whereas non-significant interactions for RU, RO, and RE may partly reflect insufficient power rather than true absence of effects.

In line with recommendations for randomised controlled trials, the primary analyses focused on between-group differences in change over time. Specifically, we tested the treatment effect using a mixed-design ANOVA with Time (pre, post) as the within-subjects factor and Group (experimental vs. control) as the between-subjects factor, with the Group × Time interaction serving as the main estimate of interest. As a robustness check, we additionally conducted ANCOVA models using post-intervention scores as dependent variables and baseline scores as covariates. Assumptions were examined using Shapiro–Wilk tests and Q–Q plots for normality and Levene’s test for homogeneity of variance. In secondary, exploratory analyses, paired-samples *t*-tests were used to describe within-leader pre–post changes in EI facets, particularly emotion regulation, although these are not interpreted as evidence of treatment effects.

To control the risk of inflated Type I error due to multiple subscale analyses, we prespecified emotion regulation (RC) as the primary outcome based on theoretical and empirical evidence linking RC most directly to stress buffering and psychological health. The remaining EI subscales (RU, RO, RE) were treated as secondary, exploratory outcomes. Because RC was defined a priori as the primary endpoint, no adjustment for multiple comparisons was applied to this test. For exploratory subscales, *p*-values are interpreted descriptively; applying Holm–Bonferroni would not change the conclusions, as none of the secondary outcomes approached significance.

Randomization was conducted after the baseline assessment using a computer-generated random sequence (simple randomization) created by the first author. Participants were allocated to either the experimental or wait-list control group in the order in which they completed the baseline questionnaire. Because the allocation process was administered electronically and no third-party randomization service was used, allocation concealment was not implemented. As the study relied on self-report questionnaires, blinding of participants and outcome assessment was not possible.

### Online Compassion and Gratitude Training

The ComGrat online self-directed training programme explored the combination of compassion and gratitude over four weeks. The programme consisted of an introductory session and twenty-eight units, with participants receiving daily text messages at 6:00 a.m. via SMS. Each message included a brief text on compassion and a loving-kindness meditation. The meditations varied across the first 14 days, after which participants could select meditations from the earlier sessions for the remaining two weeks. Every evening at 9:00 p.m., participants received a second SMS prompting them to record expressions of gratitude in a journal. This structure was designed to integrate stress-buffering practices into leaders’ daily routines in a low-threshold, scalable format that could also be applied in healthcare and other high-strain professional environments.

Emotional intelligence was measured using the Emotional Competence Questionnaire [50], which assesses four dimensions: (1) recognising and understanding one’s own feelings (RU), (2) recognising others’ feelings (RO), (3) regulating and controlling one’s own feelings (RC), and (4) expressing feelings (RE). The 62-item self-report scale demonstrated high reliability (α = 0.89–0.93; average α = 0.91). The RC subscale, reflecting emotion regulation, was treated as the most stress-proximal indicator of psychological health.

Before training, demographic data were collected (age, gender, number of children, number of employees, and years of leadership experience). After the course, participants were asked about their adherence to the programme in terms of meditation practice and gratitude journaling frequency. These adherence indicators were used descriptively to contextualise stress-related EI changes.

A mixed factorial repeated-measures ANOVA was conducted to examine changes over time. No unusual or abnormal data were detected. Levene’s test was used to assess homogeneity of variance. When assumptions were met, Tukey’s post hoc tests were applied; when violated, Holm’s method was used. The significance level for ANOVA and associated post hoc tests was set at 0.05, based on the a priori power analysis. In line with our focus on stress-management mechanisms, paired samples *t*-tests of EI subscales were emphasised as primary analyses.

## 4. Results

A total of 45 participants completed both the baseline and post-intervention surveys, and no cases were excluded from the per-protocol analysis. The sample consisted of 35 men and 10 women, aged 27 to 58 years. Random assignment resulted in comparable demographic distributions across the experimental and control groups. Completers (n = 45) and non-completers (n = 5) did not differ significantly in age, gender distribution, leadership experience, number of employees, or baseline EI dimensions (all *p* > 0.20), indicating that attrition was unlikely to bias the results.

Independent-samples *t*-tests confirmed baseline equivalence regarding age, years of leadership experience, number of employees supervised, and number of children (all *p* > 0.10). Gender distribution did not differ significantly between groups. The experimental group (n = 24) was predominantly male (n = 19), with a mean age of 45.5 years (SD = 10.4), while the control group (n = 21) had a mean age of 42.3 years (SD = 10.1). Leadership experience was similar (experimental: M = 9.75, SD = 5.01; control: M = 6.76, SD = 5.31). Although the control group showed higher variability in number of employees supervised, mean levels did not differ (Table 1).

Assumptions for parametric analysis were evaluated prior to hypothesis testing. Shapiro–Wilk tests and Q–Q plots indicated normally distributed data for all EI sub-scales (all *p* > 0.05), except for RE, which showed borderline distributional fit. Levene’s test indicated homogeneity of variance for all scales except RE. Because emotion regulation (RC)—the primary outcome—met all assumptions, analyses proceeded using parametric tests.

Adherence was assessed descriptively based on participants’ self-reported frequency of meditation practice and gratitude journaling. Because adherence responses were categorical (e.g., “daily,” “3–5 times/week,” “1–2 times/week”) rather than continuous duration or minute-based logs, the data did not permit a formal dose–response analysis. Exploratory inspection revealed no clear pattern linking adherence categories to changes in RC scores, and group-level improvements in RC were not driven by higher adherence subgroups. However, given the limited granularity of the adherence data, these observations should be interpreted cautiously.

### 4.1. Effects on Emotional Intelligence Subscales (Mixed ANOVA)

Treatment effects were examined using mixed-design ANOVAs with Time (pre, post) as the within-subject factor and Group (experimental vs. control) as the between-subject factor. The Group × Time interaction tested whether EI facets changed differentially between groups. Table 2 summarises all interaction effects. Because RC was prespecified as the primary outcome, its *p*-value is interpreted without correction for multiple comparisons. The remaining EI subscales were treated as exploratory secondary outcomes; none showed significant Group × Time interactions, and all remained non-significant after applying Holm–Bonferroni correction (all adjusted *p* > 0.20), consistent with the reported results.

#### 4.1.1. Recognising and Understanding One’s Own Feelings (RU)

The Group × Time interaction for RU was not significant, with *F*(1, 43) = 0.42, *p* = 0.52, and partial η^2^ = 0.010, indicating no differential change between the experimental and control groups. The intervention did not appear to influence participants’ ability to recognise and understand their own feelings.

#### 4.1.2. Recognising Others’ Feelings (RO)

Similarly, recognising others’ emotions showed no significant interaction effect, *F*(1, 43) = 0.55, *p* = 0.46, partial η^2^ = 0.013. Thus, the intervention did not enhance leaders’ interpersonal emotional perception within the four-week period.

#### 4.1.3. Regulation and Control of One’s Own Emotions (RC)—Primary Outcome

A significant Group × Time interaction emerged for emotion regulation (RC), *F*(1, 43) = 5.12, *p* = 0.029, partial η^2^ = 0.106, indicating that participants in the experimental group improved significantly more than those in the control group (Figure 2). The experimental group showed a substantial increase from baseline (M = 3.55, SD = 0.48) to post-intervention (M = 3.90, SD = 0.52), whereas the control group remained stable (baseline: M = 3.57, SD = 0.51; post: M = 3.62, SD = 0.56). An ANCOVA confirmed the treatment effect when adjusting for baseline scores, *F*(1, 42) = 4.87, *p* = 0.033, partial η^2^ = 0.104. This pattern supports H3 and indicates that the compassion and gratitude training specifically enhanced the EI facet most closely related to stress management.

#### 4.1.4. Expressing One’s Own Feelings (RE)

The mixed-design ANOVA revealed no significant interaction effect for RE, *F*(1, 43) = 0.28, *p* = 0.60, partial η^2^ = 0.006. Because this scale showed borderline normality, results are interpreted with caution. No evidence emerged that the intervention affected emotional expression.

The intervention produced selective improvements in emotional intelligence. Only emotion regulation (RC)—the EI facet most closely linked to stress buffering and resilience—improved significantly more in the experimental group than in the control group, as demonstrated by both mixed ANOVA and ANCOVA. No significant effects were found for RU, RO, or RE. This pattern indicates that brief online compassion and gratitude training may not enhance all aspects of EI equally but can strengthen the stress-protective mechanism of emotional regulation.

## 5. Discussion

This study examined whether a four-week, self-directed online compassion and gratitude programme could enhance leaders’ emotional intelligence (EI). The findings showed a selective improvement in emotion regulation (RC) among participants in the intervention group, while recognising one’s own emotions (RU), recognising others’ emotions (RO), and expressing emotions (RE) remained unchanged. This pattern indicates that brief contemplative practices may strengthen the regulatory component of EI but are less likely to influence perceptual or expressive facets within a short training period.

The improvement in emotion regulation aligns with prior research demonstrating that compassion- and mindfulness-based practices most strongly target regulatory processes [27]. Given that emotion regulation is a key protective mechanism linked to reduced stress reactivity, healthier coping strategies, and lower burnout risk in both organisational and healthcare settings [7], this selective change represents a meaningful outcome. Enhanced regulation may enable leaders to modulate stress responses more effectively, maintain emotional balance under pressure, and model constructive emotional behaviour within teams. In healthcare contexts, comparable improvements could plausibly support clinicians’ resilience, reduce emotional exhaustion, and contribute to higher-quality interactions with patients and colleagues.

The lack of intervention effects on RU, RO, and RE likely reflects the content of the programme, which focused primarily on intrapersonal regulatory strategies—loving-kindness meditation and gratitude journaling. These practices support emotional balance but do not directly involve emotion-perception or interpersonal expression tasks. Consistent with meta-analytic evidence [37], perceptual and expressive EI facets may require more interactive or relational training components, such as emotion-labelling exercises or interpersonal reflection. Such components may be especially important in healthcare environments, where accurate emotional perception and communication are essential for teamwork and patient care.

Methodological considerations also warrant attention. The study was powered to detect medium within-subject effects but not specifically powered for between-group interactions. The sample size therefore limited sensitivity to smaller effects, and the non-significant interactions for RU, RO, and RE may reflect insufficient statistical power rather than the absence of meaningful change. Nevertheless, the significant interaction for RC suggests that the observed improvement in emotion regulation is robust even under conservative power conditions. Moreover, because RC was prespecified as the primary outcome, and the remaining EI subscales were treated as exploratory, the selective effect is unlikely to be a Type I error.

Another limitation concerns the exclusive reliance on self-report EI measures. Self-assessments may be influenced by social desirability, self-reflection biases, or positive expectations inherent in psycho-educational interventions. Without behavioural or third-party assessments, it remains unclear whether improved self-rated regulation corresponds to observable changes in behaviour. Future studies should incorporate objective physiological indicators (e.g., cortisol, heart rate variability), behavioural emotion-regulation tasks, and 180° or 360° feedback from colleagues or employees to examine whether self-perceived changes translate into real-world outcomes.

The predominantly male sample (~75%) also limits generalisability. Lower female participation may reflect gender distributions within leadership roles in Germany and differential engagement with professional networking platforms such as LinkedIn and Xing. Although no gender differences were detected, the small female subsample limits the ability to examine gender-specific responsiveness to the intervention. Recruitment via social media platforms may have introduced additional selection bias, favouring individuals with higher digital engagement or interest in personal development.

Adherence represents a further limitation. Because practice frequency was collected in broad categories rather than continuous logs, dose–response relationships could not be analysed. Descriptive inspection suggested no consistent link between adherence categories and changes in RC, but objective monitoring (e.g., digital timestamps) is required in future trials to more precisely assess practice effects.

Finally, the use of a wait-list control group restricts interpretability. Without an active or expectation-matched control, non-specific factors such as expectancy, motivation, or general engagement cannot be ruled out. Future research should incorporate neutral or low-emotionality control activities (e.g., factual reading, passive relaxation, non-emotional journaling) to disentangle specific from non-specific intervention effects.

Despite these limitations, the findings provide preliminary evidence that brief, scalable, self-directed contemplative practices can strengthen a core stress-protective EI mechanism—emotion regulation—in leaders. This has practical implications for organisations aiming to support psychological well-being and mindful leadership with low-cost, accessible interventions. Healthcare-specific trials are now needed to determine whether similar benefits emerge among clinicians and whether improvements in regulation translate into downstream outcomes such as reduced emotional exhaustion, improved team functioning, and enhanced patient experience.

## 6. Conclusions

This study examined the effects of a four-week, self-directed online compassion and gratitude programme on leaders’ emotional intelligence (EI). While no changes were found in self- or other-oriented emotion recognition or emotional expression, the intervention produced a significant improvement in emotion regulation. Because regulation is the EI facet most strongly linked to stress buffering, resilience, and burnout prevention, this selective improvement underscores the potential of brief, low-threshold interventions to enhance psychological well-being.

These findings suggest that scalable and cost-effective online practices can strengthen emotion regulation not only in general leadership contexts but also in settings such as healthcare, where professionals routinely encounter high emotional demands and chronic stress. Improved emotion regulation may support healthier coping, reduce emotional exhaustion, and increase the capacity to provide compassionate and effective care.

Future research should extend beyond short-term self-report outcomes by testing more comprehensive interventions that combine digital delivery with interactive elements and by incorporating multi-source evaluations, such as 180° or 360° feedback. Crucially, studies are needed to determine whether improvements in emotion regulation translate into measurable outcomes in healthcare environments, including enhanced team functioning, better patient communication, and reduced burnout among clinicians.

Although the present findings stem from a cross-sector leadership sample, they lay important groundwork for healthcare-specific trials. Determining whether similar gains in emotion regulation occur among healthcare professionals—and whether these gains influence clinical performance and well-being—remains an important next step. Overall, the results contribute to the evidence that components of EI, particularly emotion regulation, are trainable even within short timeframes. By adopting accessible, evidence-based practices such as compassion and gratitude training, organisations and healthcare systems can foster psychological health, resilience, and mindful leadership in ways that benefit both professionals and the populations they serve.

## Figures and Tables

**Figure 1 healthcare-14-00012-f001:**
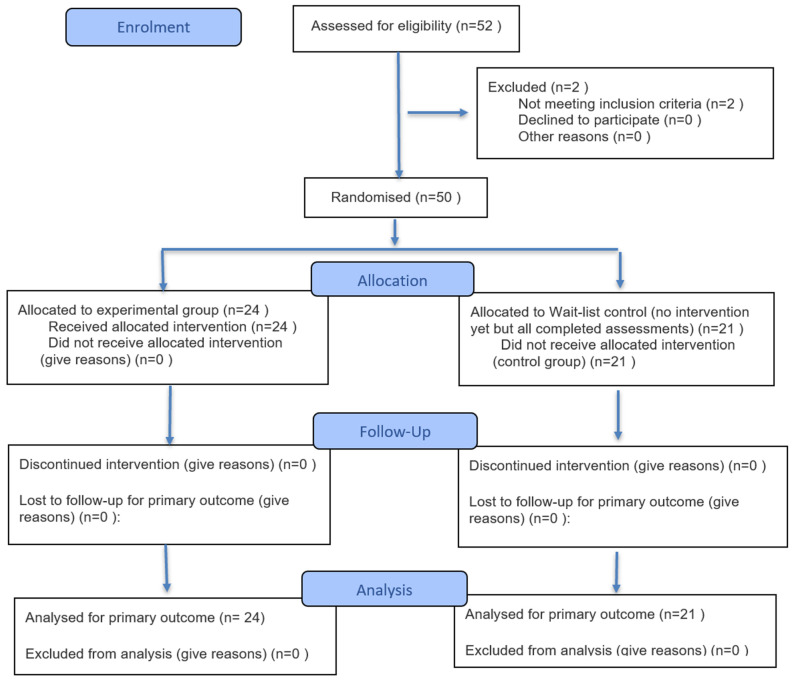
Consort Flow Diagram.

**Figure 2 healthcare-14-00012-f002:**
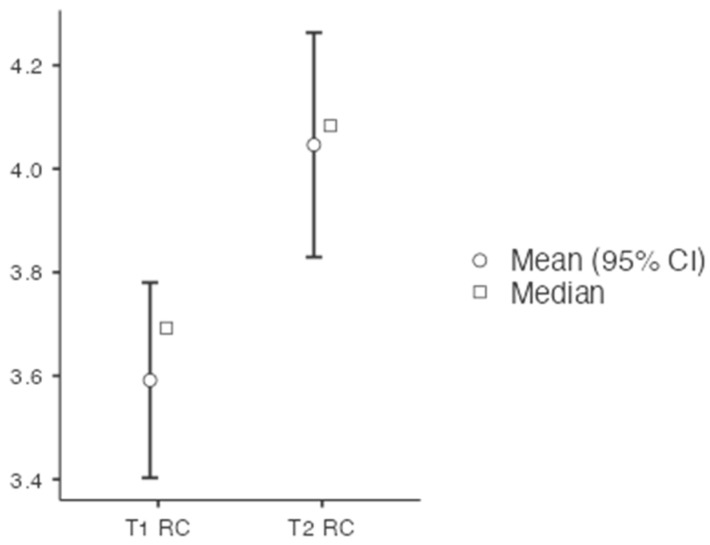
Group × Time interaction for emotion regulation (RC). Mean levels of emotion regulation (RC) at baseline and post-intervention for the experimental and control groups. Error bars represent standard errors. The experimental group showed a significant improvement compared to the control group.

**Table 1 healthcare-14-00012-t001:** Descriptive Statistics per Group (Baseline, Post) and Change Scores.

EI Subscale	Group	Baseline M (SD)	Post M (SD)	Change (Post–Pre)
RU—Recognising own feelings	Experimental	3.42 (0.52)	3.48 (0.50)	+0.06
RU—Recognising own feelings	Control	3.44 (0.49)	3.47 (0.51)	+0.03
RO—Recognising others’ feelings	Experimental	3.60 (0.55)	3.63 (0.56)	+0.03
RO—Recognising others’ feelings	Control	3.58 (0.54)	3.60 (0.55)	+0.02
RC—Emotion regulation (primary)	Experimental	3.55 (0.48)	3.90 (0.52)	+0.35
RC—Emotion regulation (primary)	Control	3.57 (0.51)	3.62 (0.56)	+0.05
RE—Emotional expression	Experimental	3.28 (0.57)	3.30 (0.58)	+0.02
RE—Emotional expression	Control	3.29 (0.55)	3.31 (0.57)	+0.02

Note. RU = recognising one’s own feelings; RO = recognising others’ feelings; RC = regulation and control of one’s own feelings; RE = expressing one’s own feelings.

**Table 2 healthcare-14-00012-t002:** Mixed-design ANOVA interaction effects (Group × Time).

EI Subscale	*F*(1, 43)	*p*	Partial η^2^	95% CI
RU—Recognising one’s own feelings	0.42	0.52	0.010	[−0.21, 0.39]
RO—Recognising others’ feelings	0.55	0.46	0.013	[−0.18, 0.42]
RC—Emotion regulation (primary)	5.12	0.029	0.106	[0.04, 0.85]
RE—Expressing feelings	0.28	0.60	0.006	[−0.23, 0.34]

Note. CI refers to the confidence interval of the interaction effect.

## Data Availability

Data available upon request due to restrictions privacy. The data presented in this study are available on request from the corresponding author. The data are not publicly available due to data protection.

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
