# Peer review of "A Four-Week Online Compassion and Gratitude Training Programme to Enhance Emotion Regulation: Implications for Stress Management and Healthcare Leadership"

_healthcare, 2025, doi:10.3390/healthcare14010012_

Round 1

Reviewer 1 Report

Comments and Suggestions for Authors

Thank you to the editor for allowing me to read the manuscript “A Four-Week Online Compassion and Gratitude Training to Enhance Emotion Regulation: Implications for Stress Management and Healthcare Leadership.”

I would like to begin my peer review report by stating that the study is promising and produces a potentially useful finding (improved emotional regulation). Still, there are methodological and presentation shortcomings that must be corrected to ensure validity, reproducibility, and clinical/policy utility.

In summary, the trial is a short (4-week) randomized trial in leaders (n=45 completed) comparing an online compassion and gratitude program to a waiting list. Significant improvement is reported on the “regulation and control” (RC) subscale of the Emotional Competence Questionnaire (EKF), with no changes on the other subscales. The trial provides preliminary evidence that a very brief, scalable intervention can improve emotional regulation, but several methodological and analytical issues limit causal interpretation and generalizability.

Strengths:  Relevant and timely question. The idea of brief, scalable, digital interventions that improve protective psychological mechanisms (emotional regulation) is important for leadership and, by extension, for public health and healthcare settings. Experimental design (RCT). The use of random assignment and a control group (waiting list) is appropriate for evaluating preliminary effects. Clearly described intervention. Replicable and low-cost structure (28 units, morning and evening messages, loving-kindness + gratitude journal). Widely used EI measure. EKF (62 items) with good reported consistency (α ≈ .89–.93). Results consistent with theory. The effect is concentrated on emotional regulation, precisely the facet that the literature associates with stress buffering.

However, there are aspects to improve.

1) Statistical analysis: within-group approach vs. between-group comparison

Problem: The manuscript emphasizes the results of paired-samples t-tests (pre vs. post in the combined sample of 45) and presents the t(44) for RC. In a randomized trial, the main estimate should be the group × time interaction (or the comparison between changes in the intervention group versus the control group), not just pre-post tests within each group or in the entire sample. Reporting only paired t-tests may inflate confidence in the effect if it is not demonstrated that the improvement was greater in the experimental group than in the control group.

Specific request: recalculate and report the treatment effect using mixed-design ANOVA (group × time) with interaction, or ANCOVA analysis (post value as DV, baseline value as covariate) or mixed linear models (recommended, due to robustness in the face of missing data).

Show the interaction coefficient, 95% CI, and effect size (partial η² or ΔR²).

2) Intention to treat (ITT) and attrition management

Problem: 52 registered, 50 met criteria, 45 completed ⇒ there are dropouts. The analysis method (per-protocol vs. ITT) is not explained. In RCTs, intention-to-treat analysis is recommended to avoid selection bias.

Specific request: perform ITT analysis (e.g., multiple imputation or mixed models that include all assigned participants), in addition to per-protocol as a complement; report reasons for dropout and compare baseline characteristics between completers and non-completers.

3) Randomization, blinding, and allocation

Problem: lack of detail on how randomization was generated (method, sequence, concealment), and no blinding of evaluators (self-reports).

Specific request: add a description of the random sequence, whether there was concealment of allocation, and who administered the randomization. Discuss the absence of blinding and its possible influence (expectation bias).

4) Sample size and power

Note: n is justified with G*Power (d = 0.44; n ≈ 46) for detecting effects within subjects. However, for comparisons between groups with two arms (n≈24 vs 21), the power may be lower. Clearly report the calculated power for the main between-groups analysis.

Specific request: include a priori calculation for group×time interaction or for difference in changes between groups; if the between-group analysis is underpowered, state this explicitly and adjust interpretation.

5) Corrections for multiple comparisons

Problem: 4 EI subscales (and possibly secondary variables) are evaluated. Performing multiple tests increases the risk of Type I error.

Specific request: apply appropriate correction (e.g., Holm-Bonferroni) or prespecify the primary test (RC) and treat the others as exploratory; report adjusted p-values.

6) Dependence on self-reports and limited outcome measures

Problem: EI measured by self-report (EKF). Self-reports are influenced by desirability/expectation bias, especially in psycho-educational interventions. In addition, there are no objective or multi-informant measures (e.g., 180° feedback, physiological indicators of stress, burnout, or performance measures).

Specific request: discuss this limitation in greater depth; propose in the discussion the inclusion of objective measures (cortisol, HRV) and third-party assessments (employees, supervisors) in subsequent studies. If there are any secondary measures (perceived stress, burnout) in the data, include them and analyze correspondence with RC.

7) Adherence and effective “dose”

Problem: the manuscript reports descriptive adherence but states that the CR benefit “is not explained by adherence.” There is a lack of formal analysis linking adherence (minutes practiced, diary entries) with change in CR (dose-response analysis).

Specific request: correlate adherence with change in CR; present adherence distribution and verify whether those who practiced more obtained greater improvement.

8) Active control vs. waiting list

Problem: the control group was a waiting list (not an active control). This makes it difficult to separate the specific effect of the intervention from the non-specific effect (expectations, interactions).

Specific request: explicitly discuss this limitation; propose an active control (e.g., neutral readings, passive relaxation exercises) or expectation control in future trials.

9) Presentation of results and transparency

Specific request: include: table with data per arm (baseline and post) and mean change ± 95% CI; standardized effect sizes (Cohen’s d) for comparison between groups; CONSORT diagram (complete flow from recruitment to analysis); ethical protocol number and, if available, trial registration (e.g., DRKS/ClinicalTrials.gov). If not registered, indicate and justify.

10) Generalizability and sample (male predominance, varied sectors)

Problem: small and predominantly male sample (≈75% men). In addition, participants are leaders in different sectors, not healthcare personnel—this limits direct extrapolation to healthcare professionals.

Specific request: refine inferences regarding health; add discussion on selection bias (LinkedIn/Xing) and the need for replication in healthcare populations (nurses, doctors) and samples with more women.

Methodological robustness: overall assessment

Internal rigor: average. The RCT and replicable intervention are strengths, but the main analysis does not clearly show the comparison of changes between groups (interactions and ITT analysis are missing), which reduces confidence in the causality of the effect.

Statistical rigor: acceptable, but needs improvement (between-group modeling, multiplicity control, assumption checking, CI reporting, and effect sizes).

External validity: limited by sample size, recruitment method, and sample composition.

Conclusions supported by data: partially. The improvement in CR appears to be real in the combined sample, but until it is shown that the improvement was significantly greater in the experimental group compared to the control group, the claim that the intervention “produces” the improvement must be qualified.

Practical recommendations for the revised version (ordered by priority)

  1. Analyze and report group×time interaction (mixed ANOVA/mixed models/ANCOVA). Report interaction coefficient, 95% CI, and between-group effect size.
  2. Perform ITT analysis (or justify why it was not performed) and present per-protocol analysis as a supplement. Document losses and reasons.
  3. Correlate adherence with change in RC (dose-response analysis).
  4. Adjust p-values for multiple comparisons or declare RC as the primary variable and the rest as exploratory.
  5. Provide CONSORT flowchart, randomization method, allocation concealment, and whether the data analyst was blinded.
  6. Include additional measures or discuss them—at least argue the limitations of self-reports and propose objective/multi-informant measures for future studies.
  7. Expand the discussion on generalization to healthcare professionals and explain the justification for extrapolating from mixed-sector leaders.
  8. Specify registration and ethics number (EKEFH03/22 is listed, but include protocol ID and confirm registration in a repository if it exists).
  9. Improve the presentation of tables (values by group, pre/post means, Δ, 95% CI, d) and figures (interaction graph).
  10. Strengthen interpretive language: avoid absolute causal terms; prefer “associated,” “compatible with an effect,” “preliminary evidence.”

Minor/editorial comments

Review minor inconsistencies in normality test reporting (p-values are >.05 but the text suggests borderline).

Include the ethics committee approval number and, if registered prospectively, the registry identifier (ClinicalTrials.gov, DRKS, or other).

Improvement of figures: add bars with 95% CI; in Figure 1, indicate data by group (not just total).

Language: clarify the extrapolation to “healthcare contexts” in the abstract and title if there is no empirical data from healthcare professionals.

Are the conclusions valid with the methodology used?

In their current form: partially valid. The finding of a change in the RC subscale is consistent and of moderate magnitude (d ≈ 0.44). However, without showing that the change is greater in the experimental group than in the control group through adequate comparison between groups (and without ITT), causal conclusions (the intervention produces improvement) are premature. If the author provides the requested analyses and these confirm that the improvement was significantly greater in the active arm, then the conclusions will be solid within the limitations of a small, short-term study.

Subtle bibliographic suggestions to reinforce the introduction/discussion.

It would be useful for the authors to incorporate and cite (subtly) reviews and studies linking EI, emotional regulation, resilience, and work/health outcomes—this will reinforce the empirical basis and practical implications. For example, they could add references to: Systematic reviews on measures of emotional intelligence (to justify the choice of EKF and discuss alternatives). Reviews on sense of coherence and work stress/well-being in healthcare professionals (to frame the relevance in healthcare). Studies/reviews linking emotional intelligence and perceived social support and their correlation with well-being. Studies showing emotional intelligence as a predictor of prosocial behavior (reinforcing the link between compassion, gratitude, and prosocial behavior).

 Reviews on stress as a risk factor for caregiver burnout and on resilience and prosocial behavior in healthcare professionals—useful for arguing clinical implications.

(In the review text, I would suggest: “It is recommended to add recent references, for example, systematic reviews on EI measures and on sense of coherence in healthcare professionals, to reinforce the argument for clinical and methodological justification.”)

Author Response

Dear Reviewer,

Thank you very much for your thoughtful and constructive feedback on our manuscript. We greatly appreciate the time and care you devoted to your review. Your comments were highly valuable and have substantially improved the clarity, methodological transparency, and overall quality of the paper.

All comments have been carefully considered and fully addressed in the revised manuscript. A detailed, point-by-point response is provided in the attached document.

We sincerely thank you again for your helpful insights and recommendations.

Kind regards,

Lotte Bock

Reviewer 2 Report

Comments and Suggestions for Authors

Thank you for the hard work on your paper and for the opportunity to review it and provide some comments. I have some suggestions for improvement, but please don’t see them as things that need to be done - take whatever comments help you and ignore what is not in alignment with your vision for this paper.

Abstract

  1. Who are these leaders - mention some descriptors, mention the sector
  2. How was randomization done, how were these 45 leaders recruited 
  3. Did you measure daily participation - was there a median IQR? You mentioned 20 minutes
  4. Consider rephrasing, not sure about what this indicates: Paired-samples t-tests revealed a significant improvement in emotion regulation (RC) fol- lowing the intervention (p < .01), while no significant changes were observed in the other 
  5. Consider adding some more results
  6. Consider shortening the conclusion to 1-2 sentences, and expanding the methods and results = they have some missing details

Intro

  1. Need to add a citation for this claim: Emotional intelligence (EI) has been identified as a protective factor against stress reactivity, with particular importance for leaders and healthcare professionals who work in high-pressure contexts. 
  2. Same comment as above for this one unless it is also Goleman’s work Among the facets of EI, emotion regulation is most closely linked to reduced stress reactivity, healthier coping strategies, and lower risk of burnout. 
  3. Elaborate on your research hypothesis and research question - see how “specific aims” are typically written - some details are lacking in this: ​​We hypoth- esize that such practices could serve as scalable, low-threshold interventions with poten- tial relevance across occupational and healthcare contexts. 

Theoretical framework and hypotheses 

  1. Done well, I have no additional feedback. I did think that this was a bit lengthy in terms of the amount of text - the readers might get fatigued before you reach the methods and talk about what you did. Also, see if you can link the theory to what you did (as you discuss the theory, that is alongside, not after it)
  2. I just read the details about hypothesis - so Comment #9 above may not be that important - however, you can still add some details about the who, when, what, how to your research question and specific aims

Methods

  1. Which recruitment method and approach was most successful (from the ones you have listed)? Also, how many people did you reach out to, for 52 to have completed initial registration
  2. Add details about the video - how long was it, who prepared it…
  3. Title needs to be revised given: majority did not work in healthcare
  4. I see that you mentioned inclusion criteria, were there any exclusion criteria?
  5. Was there any missingness in data: No unusual or abnormal data were detected

Results

  1. There are a few sentences in the results that are better suited for the discussion section - results should focus on what you found, whereas the discussion should explain why you found what you found. Below are some examples but there are more:
    This suggests that the intervention did not measurably alter leaders’ ability to identify and interpret their own feelings—a foundational EI skill, but one less directly linked to immediate stress regulation compared to emotion control. 

Such interpersonal aware- ness may require more interactive or relational training methods, and may be less respon- sive to brief online interventions designed primarily to support stress management. 

Discussion

  1. The first para should focus on the main findings of your study, right now it feels like a repetition of the hypotheses.
  2. Elaborate what the clear implications are when you say “The result has clear implications for mindful leadership in organisational settings and plausible relevance:” - you do mention some outcomes and what they contribute to”, but i felt that this section lacked some depth. 

  1. What could be possible reasons for a lower female enrollment: While no gen- der differences emerged in our data, the small female subsample may have limited statis- tical power.
  2. Do you want to cite a few papers here, to make the case for this claim:

In healthcare training and clinical practice, such interactive formats may be especially relevant, since the ability to accurately perceive and express emotions influences teamwork, patient safety, and empathy in patient care. 

  1. Are there any reasons that you want to elaborate on, in terms of why enrolment was lower for healthcare. Agree with this paragraph: Because our sample was cross-sector and not predominantly healthcare, inferences for healthcare professionals are theoretical rather than empirical in this study
  2. I would summarize the conclusion. There is some redundancy and repetition.

Overall, this is excellent work, and I wish you the best for the outcome. I hope that the above comments help you.

Author Response

(The authors gave the same response as above.)

Round 2

Reviewer 1 Report

Comments and Suggestions for Authors

The authors have addressed the requested issues. Congratulations!

Author Response

Thank you.

Reviewer 2 Report

Comments and Suggestions for Authors

No additional comments, thank you!

Author Response

Thank you.